# Separable EEG Features Induced by Timing Prediction for Active Brain-Computer Interfaces

**DOI:** 10.3390/s20123588

**Published:** 2020-06-25

**Authors:** Jiayuan Meng, Minpeng Xu, Kun Wang, Qiangfan Meng, Jin Han, Xiaolin Xiao, Shuang Liu, Dong Ming

**Affiliations:** 1College of Precision Instruments and Optoelectronics Engineering, Tianjin University, Tianjin 300000, China; mengjiayuan@tju.edu.cn (J.M.); minpeng.xu@tju.edu.cn (M.X.); flora_wk@tju.edu.cn (K.W.); jinhan@tju.edu.cn (J.H.); xiaoxiao0@tju.edu.cn (X.X.); 2Academy of Medical Engineering and Translational Medicine, Tianjin University, Tianjin 300000, China; pilgrim@tju.edu.cn (Q.M.); shuangliu@tju.edu.cn (S.L.)

**Keywords:** active brain-computer interfaces, timing prediction, discriminative canonical pattern matching (DCPM), common spatial pattern (CSP)

## Abstract

Brain–computer interfaces (BCI) have witnessed a rapid development in recent years. However, the active BCI paradigm is still underdeveloped with a lack of variety. It is imperative to adapt more voluntary mental activities for the active BCI control, which can induce separable electroencephalography (EEG) features. This study aims to demonstrate the brain function of timing prediction, i.e., the expectation of upcoming time intervals, is accessible for BCIs. Eighteen subjects were selected for this study. They were trained to have a precise idea of two sub-second time intervals, i.e., 400 ms and 600 ms, and were asked to measure a time interval of either 400 ms or 600 ms in mind after a cue onset. The EEG features induced by timing prediction were analyzed and classified using the combined discriminative canonical pattern matching and common spatial pattern. It was found that the ERPs in low-frequency (0~4 Hz) and energy in high-frequency (20~60 Hz) were separable for distinct timing predictions. The accuracy reached the highest of 93.75% with an average of 76.45% for the classification of 400 vs. 600 ms timing. This study first demonstrates that the cognitive EEG features induced by timing prediction are detectable and separable, which is feasible to be used in active BCIs controls and can broaden the category of BCIs.

## 1. Introduction

Brain–computer interfaces (BCIs) transfer the user’s intent to the computer instruction by directly decoding brain signals, which bypass the normal neural pathway including the peripheral neural and muscular systems [1,2,3]. Among all the neural imaging and recording methods, electroencephalography (EEG) has been widely applied to BCIs because of its low cost, easy to set up and high temporal resolution [4]. According to the mode of brain signal generation, EEG-BCIs are generally divided into three categories, i.e., active, reactive and passive BCIs [1,2]. This study wants to demonstrate that timing prediction can induce separable EEG features, which has the potential to work as a novel control signal for active BCIs.

Active and reactive BCIs are two typical ways to translate the brain activities into computer instructions for controlling machines. Up to now, numerous works has been done to develop the reactive BCI. In particular, the steady-state visual evoked potentials (SSVEPs) [5,6,7,8] and event-related potentials (ERPs) (such as P300) [9,10,11] are the most used brain signals. Notably, these brain signals are all triggered by external stimuli with specific patterns. That means, the reactive BCI only uses the signals generated in responses to a stimulus, and reflects users’ minds indirectly. Different from the reactive one, active BCI can access the voluntary mental activity and reflect user’s intent directly. The brain signals used for it are usually independent of external events and can be consciously controlled by the user. Thus, the active BCI could express brain’s intention in a more natural way, revealing a wide perspective in real world practice. However, the signals that can be used for active BCI are still underdeveloped with a lack of variety. Only a few cognitive EEG features have been utilized, including the motor-induced phase-unlocked EEG feature, such as the event-related desynchronization (ERD) [12,13]; the neural modulation of covert attention on sensory ERPs, such as the enhanced visual N1/P3 [14]; and error detection ERPs, such as error-related potential (ErrP) [15,16]. As a result, the brain intentions that can be decoded are limited, restricting the development of active BCIs.

It is imperative to develop new cognitive EEG features to enrich the interaction manners between the brain and computer. For this, two aspects need to be taken into further consideration. Firstly, which kind of cognitive function can express the brain’s intention in a more direct way. Secondly, whether the selected cognitive function can induce detectable and usable EEG features for BCI controls. To solve this problem, this study investigated the endogenous neural signatures triggered by timing prediction and tested the separability of corresponding EEG signatures.

Prediction is a fundamental brain function, which is crucial for our survival in the continuously changing environment. Recent neuroscience studies have demonstrated that the brain is essentially a prediction machine, which can use the prior knowledge to actively predict the external changes and guides behaviors [17,18]. Notably, during prediction processing, sensory perceptions are shaped by the prior knowledge stored in the brain, which works in the form of ‘predictive template’ [19]. Such a template is an endogenous neural signal, it can not only be independent of external stimuli, but also represent the initial intention of the brain. That means, comparing to other cognitive functions, prediction accesses the brain’s intention more directly, and is promising to provide new cognitive features for active BCI controls.

Generally, there are two categories of predictive templates, i.e., ‘what’ and ‘when’ template [20]. Neuroscience studies have observed the neural representations of both templates, but with distinct experimental instruments. Specifically, the ‘what’ template was observed with the assistance of magnetoencephalogram (MEG) [21] or high-resolution functional magnetic resonance image (fMRI) [19,22], but no EEG evidence was found. While the ‘when’ template is usually embodied as the estimation of specific time interval and could trigger specific patterns of neural oscillations in both the MEG [23,24,25,26] and EEG [27,28] studies. On that basis, this study investigated the endogenous signatures of timing prediction and tested whether distinct temporal templates can induce separable EEG features for active brain–computer interfaces. In this study, subjects were trained to construct two temporal templates, i.e., 400 ms and 600 ms, and were required to have the ability of precisely recalling them at will. Then, to verify the corresponding EEG features was useful in controlling BCIs, classification algorithms were applied to recognize the EEG features of the two predictive templates. It should be noted that this is an offline study that aims to demonstrate the recall of different predictive temporal templates is separable and can be used for BCIs.

## 2. Materials and Methods

### 2.1. Participants

Forty-five (23 females, 19–28 years old) healthy individuals, who were free from psychological or neurological diseases, participated in the experiment. All of them were right-handed and had normal or corrected to normal vision. The experimental procedures involving human subjects were approved by the Institutional Review Board at Tianjin University. All possible consequences of the experiment were explained, and the written informed consents were obtained from all the subjects. Notably, this study selected 18 subjects (10 females) according to the behavioral performance. The reason and other details were introduced in results and discussion sections.

### 2.2. Stimuli

This study embodied the ‘temporal template’ as the specific time intervals of a double-flash. It was presented by a 15*15 mm^2^ LED, which was placed at eye level 80 cm away from subjects. To obtain a precise control of time, a chronometric FPGA platform (Cyclone II: EP2C8T144C8) with a time precise of 20 ns was adapted to drive the stimuli. The detailed procedure of each experimental trial is shown in Figure 1a. To be specific, a trial was started with an auditory cue lasting for 1000ms. After the cue, there was a blank period with a duration randomly selected from 1000, 1500 and 2000 ms. Then, the double-flash emerged, in which the lasting time of each flash was 120 ms, the time-interval (TI) between the two flashes onset moments was 400, 600 or 900 ms. Despite the fact that the difference of the three intervals seems subtle, our previous study has demonstrated that subjects can successfully discriminate them [29]. Lastly, there was a blank period between 1600 and 2600 ms before the next trial. Thirty trials made up one block, in which the TI400, TI600 and TI900 double-flash were presented randomly with an equal probability. There were, in total, 12 blocks.

### 2.3. Experimental Procedure

The human brain is not very precise to millisecond timing without proper training, thus, before the formal experiment, subjects needed to be trained for about 3 days to discriminate the TI400, TI600 and TI900 double-flash. The formal experiment would not start until the subject achieved an accuracy more than 85% for at least three training blocks successively (Appendix A).

In the formal experiment, subjects were required to press the button to indicate (*i*) whether the second flash emerged at 400 ms after the first flash or not; (*ii*) whether the second flash emerged at 600 ms after the first flash or not; (*iii*) the onset of the second flash. According to the requirements, in task1, the subjects’ only predicted moment was 400 ms after the first flash, i.e., timing 400 ms (T400); in task2, 600 ms after the first flash was the only predicted moment, i.e., timing 600 ms (T600); while in task3, there was no specific expected moment in the subjects’ mind, i.e., no timing (NT). Each mental task contained four experimental blocks, and all 12 blocks were interleaved. It should be noted that in task (i) and task (ii), the predicted moment was 400 ms and 600 ms after the first flash, the subject could make a judgement by pressing the button after the predicted moment, even if the actual stimuli did not appear. Moreover, the right/left thumb was balanced across blocks. Specifically, in task1 and task2, subjects were required to press button ‘yes’ by right and ‘no’ by left thumb for two blocks, and had an exchange for the other two. In task3, they pressed the button using the right thumb for two blocks, and left for the other two.

Notably, this study only analyzed the EEG trials corresponding to the TI900 double-flash, as its second flash impinged the least disturbance on the brain expectation for both timing 400 and 600 ms. Based on data extraction, there were, in total, 40 trials for each kind of mental task; this study uses T400, T600 and NT to represent distinct mental tasks (or condition). The relationship between the expected moment and actual stimuli in the T400 and T600 conditions is shown in Figure 1b. Apparently, the endogenous brain response to expect the second flash would be earlier for T400 than T600.

### 2.4. EEG Recording and Pre-Processing

EEG signals were recorded by a Neurocan Synamps2 system at a sample rate of 1000 Hz. They were filtered by a low-pass filter at 200 Hz and a notch filter at 50 Hz. As Figure 1c showed, 64 electrodes were positioned on the scalp according to the International 10–20 system. The prefrontal lobe and the tip of the nose were selected as the ground and reference, respectively. Eye-blinks were monitored by signals recorded at FP1 and FP2. In pre-processing, the stored EEG data was first down-sampled to 200 Hz. Then, a zero-phase low-pass filter was used to cut the signals at 90 Hz, i.e., 0~90 Hz was defined as all frequency-band in this study. The band-pass filter was used to filter the signals into delta (0~4 Hz), theta (4~8 Hz), alpha (8~13 Hz), beta (15~30 Hz), and middle-gamma (45~65 Hz) bands. The frequency range of beta and mid-gamma were selected partially according to the ERSP time-frequency distribution. EEG trials were segmented from −500 to 2500 ms since the first flash onset. In event-related potential (ERP) and event-related spectral perturbation (ERSP) analysis, the period of 100 ms before the first flash was chosen as the baseline. Based on the ERP profiles, specific temporal windows were selected for amplitude and latency analysis. In the current study, the amplitude of each ERP component was calculated as the mean amplitude within the specified time window. The latency was measured as the time point before which 50% of the total component area was observed in the specified time window. Furthermore, temporal windows for classification analysis were selected according to the ERP and FDR profiles.

Additionally, the signal-to-noise ratio (SNR) was calculated and compared. The SNR in decibels (dB) could be an important indicator measuring the usability of the EEG features in BCIs; it is usually defined as the ratio of signal energy and noise energy [14]
(1)$$SNR=10\times {log}_{10}\frac{{\left(\frac{1}{N}\sum_{i=1}^{N}AMP_{i}\right)}^{2}}{\frac{1}{N}\sum_{i=1}^{N}AMP_{i}^{2}-{\left(\frac{1}{N}\sum_{i=1}^{N}AMP_{i}\right)}^{2}}$$

In formula (1), $AMP_{i}$ is the amplitude of the ith trial, and N is the number of trials.

The fisher discriminative ratio (FDR), which could reflect the separability between the data from two classes, was also calculated in this study
(2)$$FDR=\frac{{\left({\tilde{m}}_{1}-{\tilde{m}}_{2}\right)}^{2}}{{\tilde{S}}_{1}^{2}+{\tilde{S}}_{2}^{2}}$$
where ${\tilde{m}}_{k},\;{\tilde{S}}_{k}\;\left(k=1,2\right)$ respectively represent the mean value and standard deviation of the data in the kth class.

It should be noted that the above signatures, such ERP, SNR or FDR, were the grand averaged value across subjects (18 subjects) and electrodes (60 electrodes. M1, M2, CB1, CB2 were not included).

### 2.5. Feature Extraction and Pattern Classification

#### 2.5.1. Discriminative Canonical Pattern Matching (DCPM)

DCPM is a recently proposed algorithm, which achieves a perfect performance in single-trial classification of the ERP component, especially when the training set is small [14]. It consists of three parts, i.e., discriminative spatial pattern (DSP), canonical correlation analysis (CCA) and pattern matching (‘PM’ for short). Suppose $X_{k}\in R^{M\times N_{c}\times N_{t}}$ and $Y\in R^{N_{c}\times N_{t}}$ (k = 1, 2 here) represent the training and testing data set, respectively. Thereinto, M is the number of samples in the  kth class; in this study, there was 35~40 samples in each kind of data (incorrect trials were excluded).$\;N_{t}$ indicates the temporal points, 500~850 ms after the first flash was selected for DCPM classification, which contained 70 sample points.$\;N_{c}$ is the number of channels, 18 electrodes were selected for DCPM classification, including T7, C5, C3, C1, CZ, C2, C4, C6, T8, TP7, CP5, CP3, CP1, CPZ, CP2, CP4, CP6, and TP8. Before the DSP filtering, the template of the kth class was obtained by averaging the data in training set (1), as shown in Equation (3)
(3)$${\hat{X}}_{k}=\frac{1}{M}\overset{M}{\underset{m=1}{\sum }}x_{m},x_{m}\in X_{k}$$

Then, the DSP filter is constructed. The covariance matrix of ${\left[{\hat{X}}_{1},\;{\hat{X}}_{2}\right]}^{T}$ is calculated
(4)$$\sum =\left[\begin{array}{cc} \sum_{11} & \sum_{12}\\ \sum_{21} & \sum_{22} \end{array}\right]=\left[\begin{array}{cc} {\hat{X}}_{1}{\hat{X}}_{1}^{T} & {\hat{X}}_{1}{\hat{X}}_{2}^{T}\\ {\hat{X}}_{2}{\hat{X}}_{1}^{T} & {\hat{X}}_{2}{\hat{X}}_{2}^{T} \end{array}\right]$$

The $\sigma_{1}^{2}$ and $\sigma_{2}^{2}$, which are the variances of $X_{1}$ and $X_{2}$, are calculated
(5)$$\sigma_{1}^{2}=\frac{1}{M}\sum_{i=1}^{M}\left(X_{1,i}-{\hat{X}}_{1}\right){\left(X_{1,i}-{\hat{X}}_{1}\right)}^{T}\;\sigma_{2}^{2}=\frac{1}{M}\sum_{i=1}^{M}\left(X_{2,i}-{\hat{X}}_{2}\right){\left(X_{2,i}-{\hat{X}}_{2}\right)}^{T}$$

Calculate the between-class and within-class scatter matrix, i.e., $S_{B}$ and $S_{w}$:(6)$$S_{B}=\sum_{11}+\sum_{22}-\sum_{12}-\sum_{21}\;S_{w}=\sigma_{1}^{2}+\sigma_{2}^{2}$$

Then, DSP filter is adopted to construct the project matrix  U, which is the combination of a series of spatial filters. By using  U, the common mode noise could be suppressed and differences between features of the two classes could be more significant.

(7)$$S_{w}^{-1}S_{B}*U=\left[\begin{array}{ccc} \lambda_{1} & & \\ & \ddots & \\ & & \lambda_{N_{c}} \end{array}\right]*U$$

This study reduced the dimension of U and got $\hat{U}$. After DSP filtering, the templates and data became ${\hat{X}}_{1}^{T}\hat{U}$, and $Y^{T}\hat{U}.$ The second part of DCPM is the CCA, which is conducted to further find the underlying correlations between templates and testing data ($Y^{T}\hat{U}$). To do so, two projection vectors $P_{1}\in R^{N_{c}^{’}\times N_{c}^{’}}$ and $Q_{1}\in R^{N_{c}^{’}\times N_{c}^{’}}$ are needed.

(8)$$\left[P_{1},Q_{1}\right]=\begin{array}{c} argmax\\ P_{1},Q_{1} \end{array}\frac{\varepsilon \left[P_{1}^{T}{\hat{U}}^{T}{\hat{X}}_{1}Y^{T}\hat{U}Q_{1}\right]}{\sqrt{\epsilon \left[P_{1}^{T}{\hat{U}}^{T}{\hat{X}}_{1}{\hat{X}}_{1}^{T}\hat{U}Q_{1}\right]\cdot \epsilon \left[P_{1}^{T}{\hat{U}}^{T}YY^{T}\hat{U}Q_{1}\right]}}$$

In Equation (8), ε[*] is the expectation.

Then the correlations between ${\hat{X}}_{2}^{T}\hat{U}$ and $Y^{T}\hat{U}$ was calculated.

(9)$$\left[P_{2},\;Q_{2}\right]=CCA\left({\hat{X}}_{2}^{T}\hat{W},Y^{T}\hat{W}\right)$$

The third part of DCPM is pattern matching, this study mainly selected three kinds of features for DCPM analysis. The 2-D correlation coefficient, i.e., corr (*), was adopted to extract the first kind of feature for further analysis
(10)$$\rho_{1k}=corr\left({\;X\text{\textasciicircum{}}}_{k}^{T}U,Y^{T}U\right)\left(k=1,2\right)$$

And the second feature we selected for subsequent analysis is calculated as Equation (11)
(11)$$\rho_{2k}=corr\left({\hat{X}}_{k}^{T}\hat{U}Q_{k},Y^{T}\hat{U}Q_{k}\right)\left(k=1,2\right)$$

Besides, the Euclidean distance, i.e., dist(*) between ${\hat{X}}_{k}^{T}$ and $Y^{T}U$ is selected as the third kind of feature for subsequent analysis.

(12)$$\rho_{3k}=-dist\left({\hat{X}}_{k}^{T}\hat{U}Q_{k},Y^{T}\hat{U}Q_{k}\right)\left(k=1,2\right)$$

For these selected features, the larger the eigenvalue is, the closer it is to a certain kind of template.

The eigenvalue is compared, to investigate testing data is more similar to which kind of template.

(13)$${\;\rho }_{1}=\rho_{11}-\rho_{12},{\;\rho }_{2}=\rho_{21}-\rho_{22},{\;\rho }_{3}=\rho_{31}-\rho_{32}$$

Thus, the feature vector is $f_{1}={\left[\rho_{1},\;\rho_{2},\;\rho_{3}\right]}^{T}$ for the test trial.

#### 2.5.2. Common Spatial Patterns (CSP)

CSP is a spatial filter algorithm, which has been widely used in extracting ERD/ERS features. Its basic idea is to find an optimal set of spatial filters for projection, in order to concurrently maximize variance for one class and minimize variance for the other class [27]. Suppose in kth lass (*k* = 1, 2), $X_{k}\in R^{M\times N_{c}\times N_{t}}$ is the training set, and what the $N_{c}$, $N_{t}$ and M represent are the same as those in DCPM. Matrix $X_{i}\;\left(i\in X_{k}\right)$ represents the single trial EEG data of the *k*th class. After normalization, the covariance matrix of $x_{i}$ in class1 and class 2 could be respectively calculated as
(14)$$R_{1}=\frac{x_{1}x_{1}^{T}}{\mathrm{trace}\left(x_{1}x_{1}^{T}\right)},{\;R}_{2}=\frac{x_{2}x_{2}^{T}}{\mathrm{trace}\left(x_{2}x_{2}^{T}\right)}$$

The $trace\left(x_{i}x_{i}^{T}\right)$ is the sum of elements on the diagonal of a matrix. Next, the covariance matrix of mixed space covariance was calculated as Equation (15), where the ${\bar{R}}_{1}$ and ${\bar{R}}_{2}$ are the mean value of the covariances for class1 and class2 training sets. Then, an eigenvalue decomposition was conducted on *R* as Equation (16).

(15)$$U=\bar{R_{1}}+\bar{R_{2}}$$(16)$$\;R={U\lambda U}^{T}$$
where U is the eigenvector matrix of the matrix λ, and λ is a diagonal matrix constituted by the corresponding eigenvalues. Next, by arranging these eigenvalues in descending order, the whitening matrix was constructed as (17), and new matrixes for the two classes were reconstructed as (18)
(17)$$P=\sqrt{\lambda^{-1}}U^{T}$$
(18)$$S_{1}={\mathrm{PR}}_{1}P^{T}\;,{\;S}_{2}={\mathrm{PR}}_{2}P^{T}$$

Then the eigenvalue decomposition was made on the new matrices.
(19)$$S_{1}=B_{1}\lambda_{1}B_{1}^{T}\;,{\;S}_{2}=B_{2}\lambda_{2}B_{2}^{T}$$
(20)$$W_{1}=B_{1}^{T}P,W_{2}=B_{2}^{T}P$$

Here, $\;W_{1}$ and $W_{2}$ are the spatial filters we want. Suppose $X\in R^{N_{c}\times N_{t}}$ was the testing data, the new signals after spatial filtering could be calculated as Equation (21)
(21)$$Z_{1}=W_{1}^{T}X,Z_{2}=W_{2}^{T}X$$

We chose the eigenvectors corresponding to the two largest eigenvalues for each class, thus a 4 × 1 feature vector could get for each test trial:(22)$$c_{ij}=log\left\{\frac{var\left(Z_{ij}\right)}{\sum_{i=1}^{2}\sum_{j=1}^{2}var\left(Z_{ij}\right)}\right\}$$

In this study, 100~800 ms after the first flash, in total 80 points were selected for CSP classification, and the selected electrodes were same as those in the DCPM.

#### 2.5.3. The Decision-Fusion of DCPM and CSP

This study found DCPM and CSP performed better in 0~4 Hz, 20~60 Hz, respectively. Thus, we further made a decision-fusion of DCPM and CSP. Fisher discriminant analysis (FDA) was used for the decision-fusion. By finding the projection *W*, which is the weight vector maximizing the class separation, FDA projects the feature vectors $f_{1}$ (features extracted by the DCPM method) and $f_{2}$ (features extracted by the CSP method) to decision values, i.e., $F_{1}$ and $F_{2}$ as Equation (23)
(23)$$F_{1}=W_{1}^{T}f_{1}+b_{1}\;,F_{2}=W_{2}^{T}f_{2}+b_{2}$$
where $b_{1}$ and $b_{2}$ are bias constants. Then, testing data was classified by extracting the sign of the sum of $F_{1}$ and $F_{2}$, which returned 1 or −1 corresponding to T400 or T600 condition, respectively (24)
(24)$$F=sign\left(F_{1}+F_{2}\right)$$

All the classification accuracies were computed with a 10-fold cross-validation procedure, and in the classification process, a single subject was examined.

## 3. Results

### 3.1. Behavioral Performances

Behavioral performances were first analyzed to test the effectiveness of (*i*) training and (*ii*) timing prediction in formal experiment. In the training session, the reaction time was 969.17 ms, 628.08 ms, and 487.75 ms; the accuracy was 62.97%, 83.23%, and 92.56% for the 1^st^, 2^nd^, and 3^rd^ training day, respectively. Both reaction time (*F* (2,88) = 109.42, *p* < 0.001) and accuracy (*F* (2,88) = 232.367, *p* < 0.001) had significant improvements, which indicated the training was effective. As for the formal experiment, the averaged accuracy of T400 and T600 was 97.8% and 95.8%, it preliminarily indicated the subjects could make good use of the temporal templates. Additionally, for the TI900 double-flash, subjects might press a ‘No’ button before the second flash, because they could realize the time was up when they made a prediction of 400 or 600ms. For this, we eliminated the subjects who pressed the button earlier than 150 ms after the second flash, as the action of pressing the button would contaminate the EEG signals of timing prediction. Thus, 18 subjects (10 females) were left for further analyzes, and their averaged reaction time was 207.24 ms, 395.96 ms and 187.53 ms for T400, T600 and NT conditions. The following analysis were all based on the selected 18 data sets.

### 3.2. ERP Analyses

Figure 2a showed the grand-averaged ERP profiles induced by TI900 double-flash. Due to the fact that the first flash popped out randomly, the N1 component induced by the first flash should not be modulated by timing prediction. As expected, its amplitude (*F* (2,34) = 1.356, *p* = 0.271) and latency (*F* (2,34) = 0.300, *p* = 0.743) were similar among conditions. After N1, the waveforms first went downwards and then upwards, but the time for waveforms beginning to go upwards was earliest for T400, media for T600 and latest for NT conditions. Specifically, in 600~800 ms, T400 waveform had become a positive one, whereas others still went downwards, resulting in significantly different amplitude (*F* (2,34) = 7.309, *p* = 0.002; T400 vs. T600: *p* = 0.004, T400 vs. NT: *p* = 0.029, T600 vs. NT: *p* = 1.000). In 800~100 ms (Figure 2c), T600 waveform (blue) started to climb upwards, and the NT still went downwards, causing the amplitude differences (*F* (2,34) = 13.223, *p* < 0.001; T400 vs. T600: *p* = 0.028, T400 vs. NT: *p* = 0.004, T600 vs. NT: *p* = 0.048). In statistical analysis, Bonferroni correction was applied when necessary. From the above analysis, we found the contingent negative variation (CNV), i.e., the negative-going waveform, would be terminated by positive-going waveform appearing about 200 ms after the expected moment. Potentially, the positive-going waveform can be a feature reflecting the subjects had realized that time was up. Additionally, Figure 2d showed amplitude topographies averaged across 500~850 ms, which was the period selected for classification. Figure 2e showed the difference topography, indicating more differences located in the central area.

### 3.3. SNR and FDR Analyses

Figure 3a shows the SNRs of typical periods, which indicated the timing prediction can lead to SNR enhancement. Specifically, comparing to baseline (−100~0 ms), the first flash resulted in a significant SNR increasement in the N1 temporal window (135~195 ms) (T400 vs. baseline, T600 vs. baseline: *p* < 0.001; after Bonferroni correction), and then the SNR value still maintained higher than the baseline in the subsequent timing period (T400 vs. baseline, T600 vs. baseline: *p* < 0.001, after the Bonferroni correction). After the expected moment (400 ms), the SNR in T400 condition decreased gradually. Figure 3b,c shows the SNR topographies. Next, we analyzed the fisher discrimination ratio (FDR) between the T400 and T600 conditions. Two results could be observed: Firstly, FDR was larger in 600~850 ms (Figure 4b); secondly, the FDR was larger in central brain areas (Figure 4b), illustrating the central area within the period before 850 ms was more separable.

### 3.4. ERSP Analyses

Figure 5a showed the grand-averaged ERSP time-frequency distribution in T400, T600 and NT conditions. ERSP changes were observed in delta (0~4 Hz), alpha (7~15 Hz), beta (20~30 Hz), and mid-gamma (45~65 Hz) frequency-bands. As this study selected the period before 800 ms for further analyzes, we mainly compared the different-frequency band ERSPs before 800 ms. Specifically, in 0~4 Hz, T600 led to larger ERSP (Figure 2b, *F* (2,34) = 4.173, *p* = 0.024; T400 vs. T600: *p* = 0.023), which was more evident in the frontal-central area (Figure 5f). In 7~15 Hz, the N1 components induced by the first flash led to similar ERSP enhancement among conditions (*F* 2,34) = 2.987, *p* = 0.066), then ERSP suppression emerged, which was more evident in T600 and T400 than NT (Figure 2c, *F* (2,34) = 5.117, *p* = 0.011; T400 vs. NT: *p* = 0.014; T600 vs. NT: *p* = 0.040), and mainly located in the parietal-occipital brain area (Figure 5g). In the 20~30 Hz and 45~65 Hz bands, the ERSP was suppressed as well. Interestingly, such suppression emerged the earliest for T400 (at about 100 ms), media for T600 (at about 220 ms) and the latest for NT (at about 300 ms). Figure 5d,e analyzes the ERSP in 100~800 ms; both frequency-bands revealed a tendency that the suppression was larger for T400 (20~30 Hz: *F* (2,34) = 1.733, *p* = 0.288; 45~65 Hz: *F* (2,34) = 5.059, *p* = 0.012; T400 vs. T600: *p* = 0.046; T400 vs. NT: *p* = 0.031). Moreover, the two frequencies had similar distributions of ERSP suppression, both covering the whole brain. In statistical analysis, Bonferroni correction was applied when necessary.

In sum, ERSP analysis indicated the signatures induced by timing prediction were different in the distinct frequency-band, it is important to select a more suitable frequency band for classification.

### 3.5. Classification Performance

Figure 6a showed the classification results of different frequency-band EEG features using the DCPM. The averaged accuracy was 64.74%, 52.82%, 52.49%, 51.75%, 50.34%, and 63.65% for delta, theta, alpha, beta, mid-gamma, and whole-frequency (0~90 Hz) band, respectively. The corresponding standard deviation was 7.64, 6.99, 7.67, 7.02, 7.14, and 9.47. Evidently, delta band achieved the highest classification accuracy among all the conditions (*F* (5,85) = 15.369, *p* < 0.001), which was significantly higher than theta (*p* < 0.001), alpha (*p* < 0.001), beta (*p* = 0.006), and mid-gamma (*p* < 0.001); all the significance were after the Bonferroni correction. The results indicated that the low-frequency EEG features of timing prediction could be well detected by the DCPM.

Figure 6b is a classification result based on the CSP. Averaged accuracy was 53.79%, 55.38%, 60.13%, 69.31%, 68.70%, and 59.03% for delta, theta, alpha, beta, mid-gamma, and whole frequency -band, respectively. Their corresponding standard deviations were 5.32, 11.40, 10.43, 13.46, and 12.99 and 8.71. Obviously, high frequency-band EEG features achieved significantly better performance than the low-frequency (*F* (5,85) = 12.026, *p* < 0.001; delta vs. beta: *p* = 0.002; theta vs. beta: *p* = 0.005; alpha vs. beta: *p* = 0.003; delta vs. m-gamma: *P* = 0.002; theta vs. m-gamma: *p* = 0.017; and beta vs. m-gamma: *p* = 1.000;) The results indicated the high-frequency EEG features of timing prediction could be well detected by the CSP.

Furthermore, the DCPM and CSP methods were combined to improve classification accuracy. According to the aforementioned analyzes, data in the delta-band were used for the DCPM, while data in the beta and mid-gamma bands (20~60 Hz) were used for CSP in the combination method. Table 1 displayed the corresponding classification results. The combination method led to an averaged accuracy of 75.97%, with a standard deviation of 8.78. The highest and lowest accuracy was 93.75% and 63.75%, respectively; 13 subjects achieved an accuracy of more than 70%. Lastly, we compared the performance of the three classification methods. Despite the fact that the averaged CSP accuracy was higher than that of DCPM, no significant difference was found (*T* (17) = −1.386, *p* = 0.184). However, the decision-fusion method achieved much better classification performance than both the DCPM (*T* (17) = 6.086, *p* < 0.001) and CSP (*T* (17) = 2.566, *p* = 0.020). The result demonstrated that the decision-fusion of low-frequency DCPM and high-frequency CSP can improve the classification accuracy.

## 4. Discussion

Different from the reactive BCI, which depends more on the involvement of external stimuli, the active BCI reflects the user’s intent more directly via decoding their voluntary mental activities. However, few cognitive signals have been exploited for it. This study found that the brain function of timing prediction induced positive waveform in low-frequency and ERSP suppression in high- frequency, which can be decoded by the combined DCPM and CSP method, with an average accuracy of 70.06%, and the highest accuracy of 92.5%. Such detectable and separability cognitive EEG features have the potential to decode the brain’s time intention and provide new signals for the active BCIs. In the following paragraph, we would discuss the neural basis, decoding algorithm and potential disturbance regarding the above observations.

It is reasonable for timing prediction to induce positive delta waveform and beta-gamma ERSP suppression, especially in the central brain area. To be specific, delta oscillation in the central area is believed to be instrumental for reflecting many voluntary mental activities, including the perceptual judgement [20,30,31], difficulty assessment [30], target detection [31], perceptual learning [28,32], etc. In the current study, the ERP profiles started to go upwards shortly after the predicted moment, which worked as a reflection of perceptual judgement, indicating subjects have realized that time was up. As the predicted moment of T400 condition emerged earlier than that of T600, the corresponding positive-going waveform was earlier as well, resulting in the separations between the T400 and T600 profiles. Furthermore, the beta-gamma ERSP suppression observed here could also be explained by previous neuroscience studies. It is believed that the time course of beta oscillation plays a key role in maintaining predictive timing [24,25,33], and gamma oscillation, which participates in the encodings of working memory, is also involved in above maintaining process [23]. Recent studies further propose that the coupled time-structured delta waves and beta/gamma-mediated power is the key mechanism underlying timing prediction [24,26]. Additionally, the electrodes selected for classification were mainly located in central brain area, which covered motor-related regions. Such a spatial pattern accords with the fact that the motor system is involved in timing prediction [24,26], especially in the timescale less than 2000 ms [34], and the motor system often coordinates with primary sensory cortices by delta-beta coupled oscillation mentioned above [24,26]. In sum, the neural signatures observed in the current study are reasonable and can be supported by previous neuroscience studies.

For the first time, the separability of the EEG features induced by timing prediction was demonstrated, and the combined DCPM and CSP method is more suitable for decoding these EEG features. To be specific, DCPM was used to extract features in time domain. As a recently proposed method for extracting potential features [14,35,36], DCPM amplifies signal distinctions by combining the advantages of the DSP filter and CCA method. Related studies have demonstrated that DCPM can achieve better single-trial classification performance than many other feature extraction and classification methods, such as liner discriminative analysis (LDA), stepwise LDA (SWLDA) etc., especially when the data set is small (more details, see Appendix A, also see [36]). Referring to the energy feature, this study adopted the CSP, which is a traditional but robust spatial filtering method [37]. In Figure 6b, it is obvious that the classification performance of beta and mid-gamma band was much better than others, and even higher than the whole frequency-band, confirming the CSP is sensitive to frequency information [37]. Additionally, despite the fact that delta ERSP showed an evident distinction between T400 and T600, the corresponding accuracy was low. A potential explanation for it was the covariance estimation was not accurate in low-frequency, whereas the CSP mainly works by calculating the covariance estimation. Except for adopting the DCPM and CSP method separately, this study further improved the classification performance by combining the two methods at the decision level. After the decision fusion, the accuracies of 12 data sets became much higher than using the DCPM or CSP alone, especially for subject 2, 5, 7, and 8. The accuracy improvement indicates combining DCPM and CSP features at the decision level could provide more discriminative information for classification. In Appendix A, combining DCPM and CSP at the feature level was also tested; no better performance was found.

In addition, we want to discuss the potential disturbance caused by button-press. As subjects needed to make a judgement by pressing a button, someone may argue the separations of the ERP profile could alternatively be explained by button press. To exclude this confusion, we further analyzed two aspects. Firstly, we aligned the data to button-press moment, and calculated the lateralization potential, which measures the influence of button-press to ERP profiles [38,39]. In T400 and T600 condition, the lateralization potentials emerged at about 200 ms before button-press (Appendix A). That means, the influence of movement was mainly within 200 ms before button-press. Whereas the reaction time for T400, T600 and NT was 207.24 ms, 395.96 ms and 187.53 ms, respectively. Thus, the period selected for classification, i.e., 500~850 ms after the first flash, was almost not influenced by button-press. Secondly, we compared the T600 and NT profiles to exclude the confusion. If the positive-going waveform indicated button-press, the observation should be: the earlier the waveform started to go upwards, the shorter reaction time it was [38]. Nevertheless, the time when the T600 waveform started to climb positively was much earlier than that of NT, leading to significant amplitude difference (*p* = 0.048 after Bonferroni correction). Whereas the reaction time of T600 was much longer than NT (*p* < 0.001 after Bonferroni correction), which went against the above assumption. All these analyses indicate the separation between T400 and T600 profiles is attributed to timing prediction, rather than button-press. Certainly, in future study, we will simplify the experimental procedure and get rid of the button-press (Appendix A), to provide more active and natural features based on timing prediction for BCI controls.

## 5. Conclusions

This study demonstrated that the cognitive neural activities of timing prediction, especially the delta (0~4 Hz) wave in time domain and beta-gamma (20~60 Hz) power in frequency domain, are detectable and separable for different brain expectations of upcoming time intervals. For the first time, the active estimation of time was introduced to the field of BCI; it could provide new cognitive signals for the active BCIs and is promising to enrich the interaction manners between brain and computer.

## Figures and Tables

**Figure 1 sensors-20-03588-f001:**
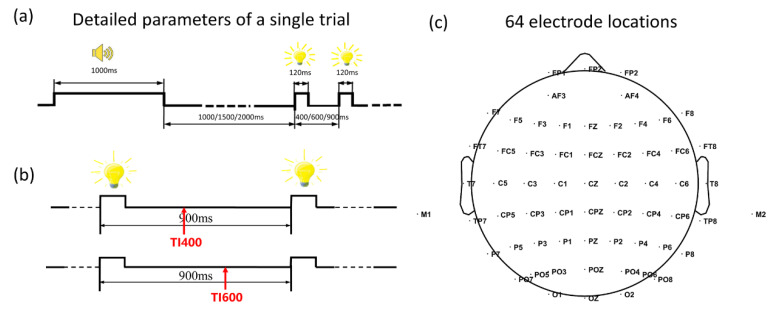
(**a**) Detailed parameters of a single trial. (**b**) Expected moment in a single trial. (**c**) 64 electrode locations.

**Figure 2 sensors-20-03588-f002:**
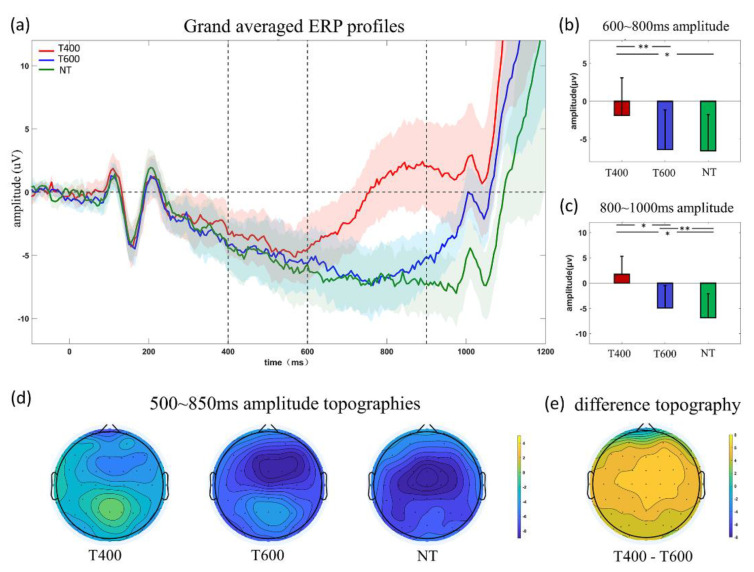
(**a**) grand-averaged ERP profiles, zero point represents the first flash onset moment; colored shadows indicate variance. The two vertical dashed lines represent the first and second flash onset moment. Amplitude comparison of (**b**) 600~800 ms; (**c**) 800~1000 ms period. (**d**) 500~850 ms amplitude topographies. (**e**) difference topographies. Statistical significance: * 0.01 < *p* < 0.05; ** 0.001 < *p* < 0.01.

**Figure 3 sensors-20-03588-f003:**
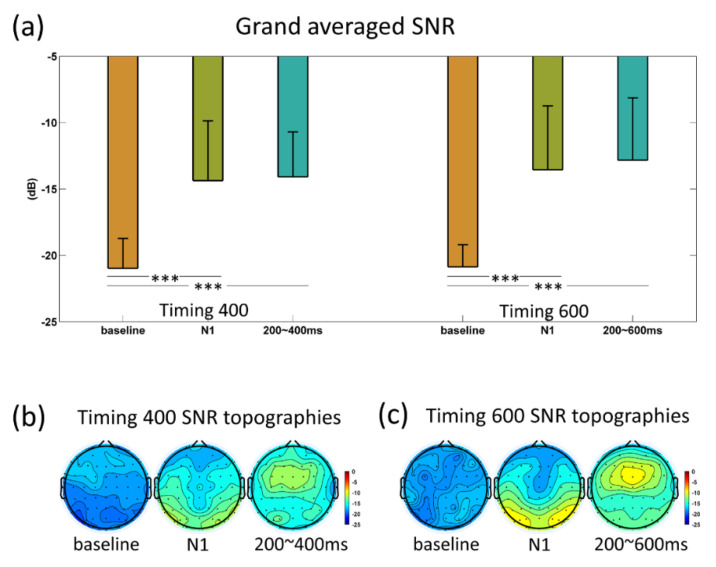
(**a**) SNR values of baseline, N1, 200~400 ms in timing 400 (left); SNR values of baseline, N1, 200~600 ms in timing 600 (right); (**b**) timing 400; (**c**) timing 600 SNR topographies. Statistical significance: *** *P* < 0.001.

**Figure 4 sensors-20-03588-f004:**
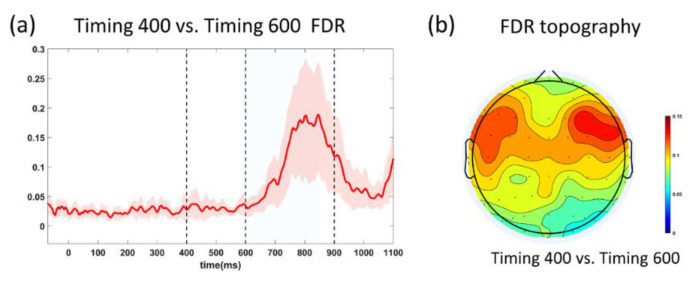
(**a**) FDR profiles; the blue shadow covering 600~850 ms (shadowed area) corresponded to the selected periods of FDR topography. The shadow in the figure indicated the standard deviation of each time point. (**b**) FDR topography of the period 600~850 ms.

**Figure 5 sensors-20-03588-f005:**
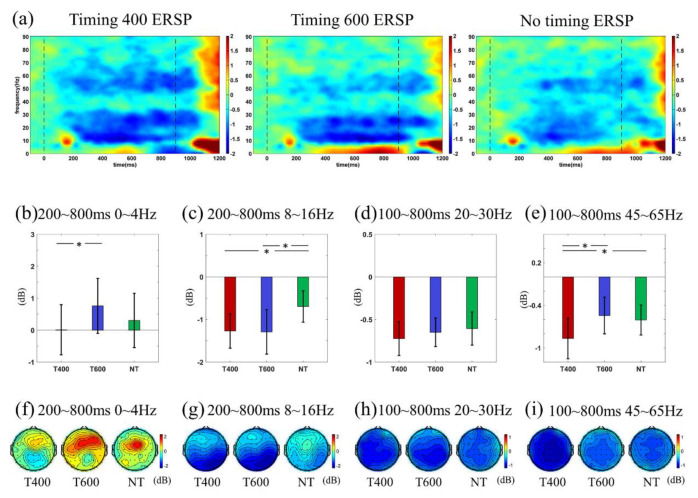
(**a**) ERSP time-frequency distribution. (**b**–**e**) ERSP values; (**f**–**i**) ERSP topographies in specific time-frequency window, which were illustrated by title. The numbers in color bars are all measured in decibel (dB). Statistical significance: * 0.01 < *p* < 0.05.

**Figure 6 sensors-20-03588-f006:**
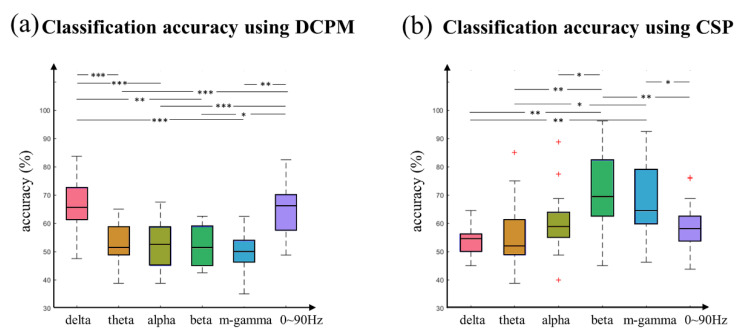
(**a**) DCPM; (**b**) CSP classification accuracy of delta, theta, alpha, beta, mid-gamma, and 0~90 frequency-band. Statistical significance: * 0.01 < *p* < 0.05; ** 0.001 < *p* < 0.01; *** *p* < 0.001.

**Table 1 sensors-20-03588-t001:** Classification accuracy comparisons.

Subject	DCPM	CSP	DCPM+CSP
	(0~4 HZ)	(20~60 HZ)	Decision-Fusion
1	62.90	82.26	82.26
2	62.50	65.00	72.50
3	75.00	40.00	71.25
4	70.00	85.00	86.25
5	78.75	55.00	88.75
6	60.00	66.25	63.75
7	61.25	63.75	75.00
8	68.75	70.00	80.00
9	67.50	72.50	76.25
10	70.00	56.25	68.75
11	66.25	71.25	73.75
12	68.75	85.00	86.25
13	48.75	72.50	71.25
14	62.50	65.00	67.50
15	70.00	92.50	93.75
16	65.00	83.75	86.25
17	47.50	72.50	67.50
18	60.00	62.50	65.00
Mean	64.74	70.06	76.45
Std	7.64	12.45	8.99

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
