# Peer review of "Separable EEG Features Induced by Timing Prediction for Active Brain-Computer Interfaces"

_sensors, 2020, doi:10.3390/s20123588_

Round 1
Reviewer 1 Report
Aiming to enrich the interaction manners between brain and computer, this paper mainly investigated the cognitive neural activities of timing prediction.
However, I have the following concerns the authors should address.
1) The related work is not well overviewed for enhancing the movitaiton of this research.
2) The contributions are not clearly stated in the introduction section.
3) The paper organization can be provided for a better readability.
4) I suggest a problem formulation or problem statement should be put in a single section.
5) I don't think the proposed method is an advanced one. How about some machine learning approaches which also might be useful for addressing this problem.
6) The presentation can be improved more.
Author Response
Dear reviewer:
Thank you for your comments and suggestions. I have revised the manuscript according to your suggestions, details are listed as follows:
- According to suggestion (1) and (3), I reorganized the introduction. In the revised introduction, the first paragraph briefly described the background about BCI, and then the pointed out the aims of this study, i.e., demonstrating that timing prediction can induce separable features for broaden the categories of active BCI. The second paragraph contained two parts: first part compared the different manners adopted by reactive and active BCI, illustrating that it is the active BCI that can express brain’s intention in a more natural way. The second part pointed out the disadvantage of active BCI: its available signals are still lack of variety, restricting the intentions that can be decoded. The third paragraph proposes that developing new cognitive EEG features is a potential approach to broaden the category of active BCI; based on this proposal, there are two aspects needing further investigation: to find out a cognitive function that can reflect brain’s intention in a relatively direct way; to test the separability of the EEG features induced by the selected cognitive function. The fourth paragraph highlighted the cognitive function of prediction can reflect brain’s intention more directly, and has the potential to broaden the signals of active BCIs. Fifth paragraph further specified prediction into timing prediction, and introduced what this study did.
- According to suggestion (2), in the revised introduction, the contribution was pointed out at the last sentence of the first paragraph, as follows: This study wants to demonstrate that timing prediction can induce separable EEG features, which has the potential to work as a novel control signal for active BCIs. Additionally, the contribution was further mentioned in the discussion section, please see the first paragraph of discussion.
- According to suggestion (4), we put the problem statement in the third paragraph. In sum, this study wants to broaden the variety of the active BCI, for the first step, two things are needed: Firstly, which kind of cognitive function can express brain’s intention in a more direct way. Secondly, whether the selected cognitive function can induce detectable and usable EEG features for BCI controls. For this, the current study investigated the endogenous neural signatures, which was independent of external stimuli, triggered by timing prediction, and tested the separability of above signatures.
- According to suggestion (5), this study added supplementary materials, which listed some results obtained by other machine learning methods. The results were also involved in discussion session.
- According to suggestion (6), the result sessions were partially rewritten (see ERP analyzes and ERSP analyzes), and discussion was also reorganized, in order to make the descriptions clearer and with less redundancy.
- In future studies, we may be able to find better classification methods. However, we think the current method, i.e., the combined DCPM and CSP method, is more suitable for the EEG features in this study. In supplementary materials, we compared the current methods with part of other machine learning methods in four aspects. Firstly, we compared the effects of classifier. In the current study, after the feature extraction, Fisher discriminative analysis (FDA) is used for distinguishing the T400 and T600 signals, we argue that the effect of FDA is similar to other classifiers, such as the support vector machine (SVM). In supplementary materials sTable1, the classification accuracies of DC+PM, DC+FDA, DC+SVM were compared. It should be noted that the DCPM method contains both feature extraction (DSP filter and CCA method, I.e., DC) and classification (Pattern Matching, PM) procedure. After the ‘DC’ feature extraction, there were three methods used for classification, i.e., PM, FDA and SVM. Obviously, different classifiers almost did not influence the classification performances of both features in time (sTable1) and frequency (sTable2) domain. Secondly, we compared the DCPM performance with traditional classification methods, such as linear discriminative analysis (LDA), stepwise LDA (SWLDA), as shown in sTable3, DCPM achieved better performance. Thirdly, there were two kinds of combination methods, i.e., combination in decision level and in feature, as shown in sTable4, the two methods had almost similar performance. Fourthly, some deep learning algorithm may have better performance, but the data set of current data was limited (40 trials at most), which obstructs the use of these algorithms. In sum, the current classification method is more suitable for the current EEG features.
Reviewer 2 Report
- The objective of the study is not clearly explained anywhere in the manuscript. This should be done in the first paragraph of the introduction.
- In the experiment, the subjects have to listen to the staring clue and then they have to respond by using right or left thumb. It is not clear how the authors were able to exclude the possibility that the differences in data did not stem from the physical response of the subject.
- In line 127 and 128 the authors have mentioned that "Notably, this study only analyzed the EEG trials corresponding to the TI900 double-flash, as its second flash would not disturb the brain expectation for both timing 400 and 600ms". This statement is in contradiction to the results provided in the manuscript.
- In figure 6, the classificaition accuracies provided are much lower than the the classification accuracies provided by the previous EEG studies. The authors should provide a clear statement for these low accuracies.
- The authors should provide more explanations about the feature extraction, whether the averaged data were used, a single subject was examined, etc.
Author Response
Answers to the suggestion from Reviewer #2:
Dear reviewer:
Thank you for your comments and suggestions. I have revised the manuscript according to your suggestions, details are listed as follows:
- According to suggestion (1), this study pointed out the contribution at the last sentence of the first paragraph in revised introduction, as follows: This study wants to demonstrate that timing prediction can induce separable EEG features, which has the potential to work as a novel control signal for active BCIs. Additionally, the introduction was reorganized, in order to enhance the motivation of this research and provide a better readability.
In the revised introduction, the first paragraph briefly described the background about BCI, and then the pointed out the aims of this study, i.e., demonstrating that timing prediction can induce separable features for broaden the categories of active BCI. The second paragraph contained two parts: first part compared the different manners adopted by reactive and active BCI, illustrating that it is the active BCI that can express brain’s intention in a more natural way. The second part pointed out the disadvantage of active BCI: its available signals are still lack of variety, restricting the intentions that can be decoded. The third paragraph proposes that developing new cognitive EEG features is a potential approach to broaden the category of active BCI; based on this proposal, there are two aspects needing further investigation: to find out a cognitive function that can reflect brain’s intention in a relatively direct way; to test the separability of the EEG features induced by the selected cognitive function. The fourth paragraph highlighted the cognitive function of prediction can reflect brain’s intention more directly, and has the potential to broaden the signals of active BCIs. Fifth paragraph further specified prediction into timing prediction, and introduced what this study did.
- According to suggestion (2), we did three analyzes to exclude the potential influence stemming from subjects’ physical response.
Firstly, this study selected the subjects, who pressed the button more than 150ms after the second flash of TI900 stimuli. Details please see result session (line 254~260): Additionally, for the TI900 double-flash, subjects might press a ‘No’ button before the second flash, because they could realize time was up when they made a prediction of 400 or 600ms. For this, we eliminated the subjects who pressed the button earlier than 150ms after the second flash, as the action of pressing button would contaminate the EEG signals of timing prediction. Thus, 18 subjects (10 females) were left for further analyses, and their averaged reaction time was 207.24ms, 395.96ms and 187.53ms for T400, T600 and NT conditions. The following analysis were all based on the selected eighteen data sets.
Secondly, in the last part of discussion, we demonstrated timing prediction is the leading cause of observed distinction, rather than the physical response. (line 422~439) However, as subjects needed to make judgement by pressing button, someone may argue the separations of ERP profile could alternatively be explained by button press. To exclude this confusion, we further analyzed two aspects. Firstly, we aligned the data to button-press moment, and calculated the lateralization potential, which measures the influence of button-press to ERP profiles. In T400 and T600 condition, the lateralization potentials emerged at about 200ms before button-press. That means, the influence of movement was mainly within 200ms before button-press. Whereas the reaction time for T400, T600 and NT was 207.24ms, 395.96ms and 187.53ms, respectively. Thus, the period selected for classification, i.e., 500~850ms after 1st flash, was almost not influenced by button-press. Secondly, we compared the T600 and NT profiles to exclude the confusion. If the positive-going waveform indicated button-press, the observation should be: the earlier the waveform started to go upwards, the shorter reaction time it was. Nevertheless, the time, when the T600 waveform started to climb positively, was much earlier than that of NT, leading to significant amplitude difference (P=0.048 after Bonferroni correction). Whereas the reaction time of T600 was much longer than NT (P<0.001 after Bonferroni correction), which went against the above assumption. All these analyzes indicate the separation between T400 and T600 profiles is attributed to timing prediction, rather than button-press. Certainly, in future study, we will simplify experimental procedure and get rid of the button-press, to provide more active and natural features based on timing prediction for BCI controls.
Thirdly, the variations of physical responses could essentially be attributed to distinct timing prediction, which did not influence the distinction between T400 and T600 timing predictions.
- This sentence has been changed into “Notably, this study only analyzed the EEG trials corresponding to the TI900 double-flash, as its second flash impinged the least disturbance on the brain expectation for both timing 400 and 600ms”.
- As to suggestion (4), we want to explain two points. Firstly, figure 6 mainly aimed to compared the classification performances of distinct frequency-bands. It is reasonable for some unsuitable frequency-bands to reveal lower classification accuracy. Secondly, the current study mainly wants to verify the separability of the EEG features induced by timing prediction, which could provide an approach for decoding the brain’s time intention, and can be improved in future study. However, previous studies did not decode such signals.
- According to suggestion (5), we added more details in the methods session. For example: (line 160~161): It should be noted that the above signatures, such ERP, SNR or FDR, were the grand averaged value across subjects (18 subjects) and electrodes (60 electrodes). (line 250~251): All the classification accuracies were computed with a ten-fold cross-validation procedure. And in classification process, a single subject was examined.
Reviewer 3 Report
The introduction of this paper is well written, to some extent method is also well-written with some flaws. The representation of results is fine but after the results section, it becomes very difficult to read the paper.
It is not clear where in the experimental design the time for reacting to the flash after 400 ms or 600ms was provided and how long was it. The authors have written it was different and have given an average reaction time but this time period is not shown in the experimental design.
Although the figures are great and show the results in a nice manner its description does not do justice to the results presented and it is at a time very difficult to understand. Please rewrite the result and discussion section to reduce redundancy.
Some minor comments:
Line 105 -106: Authors write, “our previous study has demonstrated…”. Could the authors give reference to their previous studies? I have seen that there is the reference in the discussion section but it should be cited where it is reference first.
Line 113: “Due to the human brain is not” the sentence beginning is not correct. Please correct it.
Line 115-116: authors write, “ until the subject achieved an accuracy more than 85% for at least three training blocks successively”. Can the author provide data on this? How many blocks of training were required to achieve this phase?
Line 150: The abbreviation should be expanded on its first use.
Author Response
Answers to the suggestions from Reviewer #3:
Dear reviewer:
Thank you for your comments and suggestions. I have revised the manuscript according to your suggestions, details are listed as follows:
- According to the suggestion, we added an explanation in experimental design, please see line 120~122: It should be noted that in task (i) and task (ii), the predicted moment was 400ms and 600ms after 1st flash, subject could make a judge by pressing button after the predicted moment, even if the actual stimuli did not appear. As to the length, it was just a keystroke.
That means, subjects could press the button to make judgement as long as the predicted moment was passed.
- According to suggestion (2). We rewrite the result and discussion section to reduce redundancy. To be specific, in result session, we mainly rewrite the description about ERP and ERSP, please see line 275~291 (ERP) and line 313~330 (ERSP). Some small changes were also conducted in the contents of behavioral performance, SNR analysis.
In the revised discussion, we first summarized the contribution and observation of this study. Then, the results were discussed from three aspects. Firstly (line 379~397), the neural basis of positive delta waveform and beta/gamma ERSP suppression. It indicated neural signatures observed in this study can be supported by previous neuroscience studies. Secondly (line 398~418), the combined DCPM and CSP method is more suitable for decoding the EEG features in the current study, the content included the advantages of DCPM, the classification performance of CSP, the combined method had better performance, etc. Lastly (line 419~437), we discussed the potential disturbance and what should be done in future study.
Answers to some minor comments:
- we cited our previous study where it is referenced first, please see line 103
- we changed the incorrect sentence into “The human brain is not very precise to millisecond timing without proper training, thus, before formal experiment, subjects needed to be trained for about three days, to discriminate the TI400, TI600 and TI900 double-flashes.”
- According to the suggestion, we added the behavioral data in supplementary materials (s1). As the ability of perceiving timing intervals varied with subjects, so the training blocks for each subject were different. In each training day, the subjects with better time perception, only did about 5 training blocks; whereas the ones who was insensitive to perceive the timing intervals, needed to train for 10~20 blocks. For this reason, the current study mainly calculated the averaged behavioral result of each training day. As to the sentence “at least three training blocks successively”, it was monitored by the experimenter during training.
- The abbreviation, i.e., SNR, has been expanded on its first use.
Round 2
Reviewer 1 Report
I think the authors have address the comments that I concerned. It can be considered to be "Accept" in this form.
Author Response
Thank you very much for your comments and suggestions.
Reviewer 3 Report
The authors have done a great job in answering all the comments and revising the paper.
Please fix these some minor issues:
Line 41 to 42: Authors write, "BCI can only 42 read users’ minds indirectly", BCI cannot read the user's mind. BCI only uses the signals generated in response to a stimulus. Please correct this sentence.
Line 51: Please correct the spelling of word, "intensions".
Line 53 to 60: The newly added paragraph is good but it is difficult, please change this paragraph and make it more readable.
Line 98: please write, "is shown".
Line 95 to 105: Now the stimuli section explains the paradigm well but there are still some grammatical mistakes. Please check.
Author Response
Dear reviewer:
Thank you very much for your suggestions. I corrected the mistakes as follow:
1) Line 41 to 42: I corrected this sentence into: That means, the reactive BCI only uses the signals generated in response to a stimulus, and reflects users’ minds indirectly.
2) Line 51: I corrected the spelling of word, "intentions".
3) Line 53 to 60: I changed the paragraph into: It is imperative to develop new cognitive EEG features to enrich the interaction manners between the brain and computer. For this, two aspects need to be taken into further consideration. Firstly, which kind of cognitive function can express brain’s intention in a more direct way. Secondly, whether the selected cognitive function can induce detectable and usable EEG features for BCI controls. To solve this problem, this study investigated the endogenous neural signatures triggered by timing prediction, and tested the separability of corresponding EEG signatures.
4) Line 98: I have corrected it, "is shown".
5) Line 95 to 105: I checked the paragraph, and the words that have been corrected were shown in BOLD type:
- This study embodied the ‘temporal template’ as the specific time intervals of a double-flash. It was presented by a 15*15 mm2 LED, which was placed at the eye level 80cm away from subjects. To obtain a precise control of time, a chronometric FPGA platform (Cyclone â…¡: EP2C8T144C8) with a time precise of 20 ns was adapted to drive the stimuli. The detailed procedure of each experimental trial is shown in Figure 1(a). To be specific, a trial was started with an auditory cue lasting for 1000ms. After the cue, there was a blank period with a duration randomly selected from 1000, 1500 and 2000ms. Then, the double-flash emerged, in which the lasting time of each flash was 120ms, the time-interval (TI) between the two flashes onset moments was 400, 600 or 900ms. Despite the difference of the three intervals seems subtle, our previous study has demonstrated that subjects can successfully discriminate them [31]. Lastly, there was a blank period between 1600 and 2600ms before the next trial. Thirty trials made up one block, in which the TI400, TI600 and TI900 double-flash were presented randomly with an equal probability. There were totally twelve blocks.